# DPPH Radical Scavenging Activity of New Phenolics from the Fermentation Broth of Mushroom *Morehella importuna*

**DOI:** 10.3390/molecules28124760

**Published:** 2023-06-14

**Authors:** Feifei Wang, Jie Tan, Ruixiang Jiang, Feifei Li, Renqing Zheng, Linjun Yu, Lianzhong Luo, Yongbiao Zheng

**Affiliations:** 1Engineering Research Centre of Industrial Microbiology, Ministry of Education, College of Life Sciences, Fujian Normal University, Fuzhou 350117, China; wangfeifei1119@163.com (F.W.); qsx20221239@student.fjnu.edu.cn (J.T.); r990917@163.com (R.J.); lifeifei0227@126.com (F.L.); qsz20221630@student.fjnu.edu.cn (R.Z.); yulj113@fjnu.edu.cn (L.Y.); 2Provincial University Key Laboratory of Cellular Stress Response and Metabolic Regulation, College of Life Sciences, Fujian Normal University, Fuzhou 350117, China; 3Engineering Research Center of Marine Biopharmaceutical Resource, Xiamen Medical College, Xiamen 361023, China

**Keywords:** *Morehella importuna*, isobenzofuranone, orsellinaldehyde, antioxidant, microbial fermentation

## Abstract

In recent years, wild morel mushroom species have begun to be widely cultivated in China due to their high edible and medicinal values. To parse the medicinal ingredients, we employed the technique of liquid-submerged fermentation to investigate the secondary metabolites of *Morehella importuna*. Two new natural isobenzofuranone derivatives (**1**–**2**) and one new orsellinaldehyde derivative (**3**), together with seven known compounds, including one *o*-orsellinaldehyde (**4**), phenylacetic acid (**5**), benzoic acid (**6**), 4-hydroxy-phenylacetic acid (**7**), 3,5-dihydroxybenzoic acid (**8**), *N*,*N*′-pentane-1,5-diyldiacetamide (**9**), and 1H-pyrrole-2-carboxylic acid (**10**), were obtained from the fermented broth of *M. importuna*. Their structures were determined according to the data of NMR, HR Q-TOF MS, IR, UV, optical activity, and single-crystal X-ray crystallography. TLC-bioautography displayed that these compounds possess significant antioxidant activity with the half DPPH free radical scavenging concentration of 1.79 (**1**), 4.10 (**2**), 4.28 (**4**), 2.45 (**5**), 4.40 (**7**), 1.73 (**8**), and 6.00 (**10**) mM. The experimental results would shed light on the medicinal value of *M. importuna* for its abundant antioxidants.

## 1. Introduction

The genera Morchella (Morel), which includes these species *M. importuna*, *M. esculenta*, *M. sextalata*, and *M.* eximia, are edible medicinal fungi of the phylum Ascomycota with high gastronomic quality and potential therapeutic use [1]. The fruiting body of these species contained abundant bioactivities substances [2]. It was reported that the aqueous extract from *M. importuna* possessed antileishmanial activity, a novel peptide from *M. importuna* could induce apoptosis in HeLa cells, phenolic compounds from *M. esculenta* with high antioxidant activity [3], and sterols and trilinolein showed significant inhibition of NF-κB activation [4]. However, the wild morel is rare and could not be artificially cultivated until recent years. To utilize the natural resources of this genus mushroom, mycelia prepared by microbial fermentation technology were an alternative product for the investigation of its chemical constitution with pharmaceutical potential. Some reports provided the convincing truth that mycelia and other products prepared from submerged culture are a valid alternative to the fruiting body, such as *M. esculenta* [5]. Microbial fermentation was generally used to explore the metabolites of some wild and rare mushrooms in our lab. In this paper, we mainly focus on the isolation, chemical structures, and antioxidant properties of the low-molecule metabolites of *M. importuna* by microbial fermentation. As a result, two new isobenzofuranone derivatives and one new orsellinaldehyde derivative, together with five phenolics and two *N*-contained metabolites hexamethylene bisacetamide and 1*H*-pyrrole-2-carboxylic acid, were isolated from the fermentation broth. These natural metabolites show significant antioxidant activity.

## 2. Results

### 2.1. Structure Identification

Ten compounds were purified from the organic extract prepared from the fermentation broth of *M. importuna* as reported in detail in the experimental part.

Inspection of the spectral data of compounds **1**–**2** (Table 1) indicated that they are derivatives of isobenzofuranone, which possess one phenyl group substituted by two meta phenolic hydroxyls according to two meta-coupled aromatic protons (Table 1). Compound **1** was isolated as a white amorphous substance with ${\left[a\right]}_{D}^{20}$ − 23.0 (c 0.36, MeOH) and UV (methanol) λ_max_, nm: 209. The molecular formula was determined to be C_9_H_8_O_5_ by High-resolution Quadrupole Time-of-Flight Mass Spectrometry (HR Q-TOF MS) at 197.0441 for [M + H]^+^ (calculated for C_9_H_9_O_5_, 197.0450). The IR spectra showed the absorptions for hydroxyl (3147 cm^−1^), ester carbonyl, and aromatic unsaturated double bond (1623 cm^−1^). The ^13^C NMR data revealed six carbon signals (δ 103.1, 109.6, 123.4, 131.3, 155.5, and 162.5) for one phenyl group, which was substituted by two meta phenolic hydroxyls according to the chemical shift and coupling constant of two protons [δ 6.74 (d, *J* = 2.0 Hz) and 6.68 (d, *J* = 2.0 Hz)]. The key heteronuclear multiple-bond correlation (HMBC) spectrum of **1** exhibited correlations from H-7 to C-1, C-5, C-6, and C-3a, H-5 to C-4, C-6, C-7, and C-3a, and H-3 to C-1 and C-8 and allowed the establishment of isobenzofuranone substituted with two hydroxyls and one methoxyl groups in **1**. The planar structure of **1** was characterized as one new derivative of isobenzofuranone, namely 4,6-dihydroxy-3-methoxyisobenzofuran-1(3H)-one. The absolute configuration of C-3 was assigned by the optical rotation test. Rubralide A [6] isolated from *Penicillium rubrum* showed the dextral optical rotation with the value of ${\left[a\right]}_{D}^{20}$ + 4.0 (c 0.5, EtOH), whose only one chiral carbon has an (*R*)-configuration at C-3. The sinistral rotation value of ${\left[a\right]}_{D}^{20}$ − 23.0 (c 0.36, MeOH) of **1**, indicated the (*S*)-configuration at C-3. Therefore, compound **1** was concluded as (*S*)-4,6-dihydroxy-3-methoxyisobenzofuran-1(3H)-one. Compound **2** was isolated as white amorphous substance with the UV (methanol) λ_max_, nm: 224. The molecular formula of **2** was determined to be C_9_H_8_O_5_ by HR Q-TOF MS at 197.0443 for [M + H]^+^ (calculated for C_9_H_9_O_5_, 197.0450). The IR spectra showed the absorptions for hydroxyl (3312 cm^−1^), ester carbonyl (1684 cm^−1^), and aromatic unsaturated double bond (1637 cm^−1^). The NMR spectrum of **2** was similar to those of **1**, except for the presence of the proton at δH 10.33 (H-3) and carbon signal at δC 196.6 (C-3) (Table 1). The downfield chemical shift of H-3 and C-3 indicated the presence of a formaldehyde group in compound **2**, compared with the hemiacetal group at same position in compound **1**. HMBC correlations from H-7 to C-5, C-6, and C-3a, H-5 to C-4, C-6, and C-3a, H-3 to C-1 and C-5, and H-8 to C-1 were observed in HMBC spectrum of compound **2**, indicating the opening of the furanone ring of an isobenzofuranone skeleton. Compound **2** was deduced as methyl 2-formyl-3,5-dihydroxybenzoate according to the IUPAC system, a hydrolysis product of **1** (Figure 1). Compound **2** was often used as the building block for the synthetic medicinal chemicals [7] and reported as a natural product for the first time in this paper. Isobenzofuranone is also known as phthalide and is widely distributed in microbes and plants [8]. In recent years, many natural molecules with this skeleton were discovered and shown diverse biological activities, such as antibacterial [9], antifungal [10], insecticidal [11], cytotoxic [12], anticyclooxygenase-2 [13], anti-acetylcholinesterase [14], α-glucosidase inhibitory effect [15], anticoagulation [16], hepatoprotective [17], and neurodegenerative prevention [18], etc. So, phthalide analogs exhibit a potential pharmacological value. Except for *Hericium erinaceus* [19,20] and *Pleurotus djamor* [21], few reports refer to phthalide-related metabolites isolated from mushrooms. New isobenzofuranone derivative **1** from mushroom *M. importuna* merits further investigation for its bioactivities.

Compound **4** was isolated as a white powder. The ^13^C NMR data (recorded in (CD_3_)_2_OD, 500 M) revealed eight carbon signals for one methyl (δ_C_ 18.2), six *sp*^2^ carbons (δ_C_ 113.2, 167.1, 101.4, 166.5, 111.6, 146.0), and one carbonyl carbon (δ_C_ 194.2) in compound **4**. Six protons’ peaks for δ_H_ 10.09 (s, 1H), δ_H_ 6.30 (dd, *J* = 2.3, 0.9 Hz, 1H), 6.17 (d, *J* = 2.3 Hz, 1H) and δ_H_ 2.53 (s, 3H) were observed in ^1^H NMR spectra of **4**. Based on the above ^1^H and ^13^C NMR data, **4** was identified as 2,4-dihydroxy-6-methylbenzaldehyde, namely *o*-orsellinaldehyde, which was confirmed by X-ray diffraction (Figure 1). Crystallographic data (CCDC 2214836) for **4**: C_8_H_8_O_3_, monoclinic, space group P21/c, *a* = 7.4221(4) Å, *b* = 13.1943(8) Å, *c* = 7.1941(4) Å, *α* = 90°, *β* = 90.425 (5)°, *γ* = 90°, *V* = 704.50 (7) Å3, *Z* = 4, *Dc* = 1.434 g·cm^−3^, *F*(000) = 320, 16,396 reflections measured, 1218 unique (*R_int_* = 0.0862) which were used in all calculations. The final ω*R*_2_ was 0.1361 (all data) and *R*_1_ was 0.0495 (*I* ≥ 2σ (*I*)).

Compound **3** was isolated as a white powder with UV (methanol) λ_max_, nm: 202. The molecular formula of compound **3** was determined to be C_9_H_10_O_4_ based on the HR Q-TOF MS peak at *m/z*: 261.1823 (calculated for C_9_H_10_O_4_Na, 261.1830) and NMR data. The IR spectra showed the absorptions for hydroxyl (3433 cm^−1^) and an aromatic unsaturated double bond (1630 cm^−1^). Additionally, ^1^H NMR [(CD_3_)_2_OD, 500 M)]: 6.25 (s, 1H, H-5), 10.09 (s, 1H, H-7), 2.51 (s, 3H, H-8), 3.70 (s, 3H, H-9). Moreover, ^13^C NMR [(CD_3_)_2_OD, 125 M)]:113.1 (C-1), 135.9 (C-2), 140.5 (C-3), 160.1 (C-4), 100.9 (C-5), 162.3 (C-6), 194.6 (C-7), 10.5 (C-8), 61.1 (C-9). HMBC correlations from H_3_-8 to C-1/2/3, from H-7 to C-6, from H-5 to C-1/3/4/6, and from H_3_-9 to C-3 could be observed in HMBC spectra. Compound **3** was a new derivative of **4** and was identified as 4,6-dihydroxy-3-methoxy-2-methylbenzaldehyde (Figure 1). *o*-Orsellinaldehyde was reported as a bioactive metabolite produced by the mushroom *Grifola frondosa* with a selective cytotoxic effect and anti-inflammatory and pro-apoptotic properties [22,23], and *Phlebiopsis gigantea* with antifungal activity [24].

Analysis of ^1^H and ^13^C NMR spectra data (Appendix A), compounds **5**–**8** were identified as four aromatic carboxylic acids containing the carboxylic group and the benzene ring, named as 2-phenylacetic acid, benzoic acid, 2-(4-hydroxyphenyl) acetic acid, 3,5-dihydroxybenzoic acid (Figure 1). The experimental results indicated *M. importuna* could yield a wide variety of phenolics. Phenolics are the vital chemical ingredient of some culinary and medicinal mushroom, such as *Phellinus pini* [25], *Flammulina velutipes* [26], *Inonotus obliquus* [27], *Ganoderma lucidum* [28], *Pleurotus citrinopileatus* [29], *Hypsizygus marmoreus* [30], *Porodaedalea chrysoloma* [31], *Antrodia cinnamomea* [32], and *Tuber indicum* [33], etc. Phenolics exhibit antioxidant activity and provide health benefits.

Compound **9** was isolated as a white powder with ESI-MS *m/z*: 187 [M + H]^+^. Additionally, ^1^H NMR (600 MHz, CDCl_3_) δ: 5.92 (s, 2H, 2 × NH), 3.23 (m, 4H, H-1, 5), 1.52 (m, 4H, H-2, 4), 1.36 (m, 2H, H-3), 1.98 (s, 6H, 2 × CH3). Furthermore, ^13^C NMR (151 MHz, CDCl_3_) δ: 170.6 (2 × CO), 39.3 (C-1, 5), 29.0 (C-2, 4), 23.8 (C-3), 23.4 (2 × CH_3_). Compared with the literature NMR data, **9** was identified as *N*,*N*’-pentane-1, 5-diyldiacetamide which was isolated as a natural product from *Blaps japonensis* [34]. The hybrid polar-planar compound **9** is a hexamethylene bisacetamide (HMBA) analog. It was reported HMBA was a potent inducer of erythroleukemic differentiation [35] and an inhibitor of vascular smooth muscle cell proliferation [36]. HMBA and its analogs induce hexamethylene bis-acetamide inducible protein I (HEXIM1) expression in cancer cells and achieve its biological activity. Although the molecule failed at Phase II clinical trial because of the dose-dependent toxicity, HMBA analogs showed the pharmaceutical potential.

Compound **10** was isolated as a white powder. Additionally, ^1^H NMR (600 MHz, CDCl3) δ: 7.34 (d, *J* = 3.5 Hz, 1H, H-3), 6.57 (dd, *J* = 3.5, 1.7 Hz, 1H, H-4), 7.65 (d, *J* = 1.7 Hz, 1H, H-5). Moreover, ^13^C NMR (151 MHz, CDCl3) δ: 143.9 (C-2), 120.3 (C-3), 112.4 (C-4), 147.6 (C-5), 163.3 (COOH). Compound **10** was identified as 1H-pyrrole-2-carboxylic acid (PCA) and further confirmed by X-ray diffraction (Figure 1). Crystallographic data (CCDC 2214809) for **10**: C_5_H_5_O_2_, brown crystal, monoclinic, space group C2/c, *a* = 13.2599(14) 10^−10^ m, *b* = 5.0253(5) 10^−10^ m, *c* = 14.8921(16) 10^−10^ m, *α* = 90°, *β* = 99.199(11)°, *γ* = 90°, *V* = 979.57(18) 10^−30^ m^3^, *Z* = 4, *Dc* = 1.507 g∙cm^−3^, *μ* (Mo-Kα) = 0.119 mm^−1^, *F*(000) = 464.3, 2112 reflections measured, 1025 unique (*R_int_* = 0.0128) which were used in all calculations. The final *ωR*_2_ was 0.1426 (all data) and *R*_1_ was 0.0366 (*I* ≥ 2σ (I)). PCA was previously reported as an antimicrobial natural product obtained from bacteria, the sponge *Agelas nakamurai*, and the Chinese herb *Pseudostellaria* heterophylla [37,38]. It was discovered for the first time as a mushroom metabolite in this paper.

### 2.2. DPPH Free Radical Scavenging Activity

In the TLC-bioautography experiments, the various degrees of white spot observed in the different lanes indicated that these compounds **1**, **2**, **4**, **5**, **7**, **8**, and **10** possessed significant antioxidant activity (Figure 2). Their antioxidant activities were further assessed by the Brand-Williams’ method. DPPH radical scavenging rate of these compounds with the different doses were shown in Figure 3. The half DPPH scavenging concentration (SC_50_) of these compounds were 1.79 (**1**), 4.10 (**2**), 4.28 (**4**), 2.45 (**5**), 4.40 (**7**), 1.73 (**8**), and 6.00 (**10**) mM, compared with the value of 0.216 mM of vitamin C, which were estimated by using Probit analysis in SPSS 18.0 with *p* value less than 0.01 except for 0.064 for **2**. Most of these compounds displayed significant anti-DPPH radical potency. These data suggest that *M. importuna* could be used as a source of abundant natural antioxidants.

The DPPH radical (DPPH•) bearing a stable unpaired electron is regarded as one convenient method for determining the antioxidant activity of a wide variety of organic molecules [39]. The intracellular free radicals were believed to be involved in a diverse range of diseases and accelerating the aging process [40]. Antioxidants with the ability to scavenge free radicals and reduce oxidative stress are reported to be consecrated with all types of pharmacological applications [41]. In recent years, the antioxidant substance obtained from mushrooms were discovered to confer a beneficial effect on human health, such as antidiabetic and antihyperglycemic [42], neuroprotective [43], hepatoprotective [44], anti-inflammatory [45], immunomodulatory [46], anti-aging [47], anticancer [48], antimicrobial [49], anti-melanogenic [50] and hypopigmentation [51], anticoagulant [52], nephroprotective [53], and hyperuricemia treatment [54], etc. The abundant natural antioxidants of *M. importuna* will lay the foundation for its medicinal benefits.

## 3. Materials and Methods

### 3.1. General Experimental Procedures

NMR spectra were acquired on a Bruker AVANCE Ⅲ 500 spectrometer operating at 500/125 MHz and a Bruker AVANCE NEO 600 spectrometer operating at 600/150 MHz. UV spectra were recorded on a Shimadzu UV-2401PC spectrophotometer in nm (λ_max_). IR spectra were measured with a Bruker Vertex 70 FT-IR spectrophotometer with KBr cells in cm^–1^. Optical rotations were measured on a Rudolph Research Analytical polarimeter (Autopol VI). HR QTOF MS spectra were recorded on an Agillent 6520 mass spectrometer in the positive mode (4.0 KV) over the mass range *m*/*z* 200 to 800. X-ray single diffraction was performed on an Oxford Gemini S Ultra diffractor. SpectraMax^®^ i3x multi-mode microplate reader (Molecular Devices) was used to measure the absorbance. Column chromatography was accomplished over silica gel (Qingdao Marine Chemical Company, Qingdao, China), reverse phase octadecyl-silica RP-C18 (Merck, Darmstadt, Germany), and Sephadex LH-20 (Amersham Biosciences, Piscataway, NJ, USA). Medium-pressure liquid chromatography (MPLC) was performed using a Quiksep-50IID (H&E, Beijing, China). Thin layer chromatography (TLC) was performed on the precoated silica gel GF254 plates (Qingdao Marine Chemical Company, Qingdao, China). Organic solvents used were from Sino-pharm Chemical Reagent Co., Ltd. (Shanghai, China). Additionally, 2,2-diphenyl-1-picrylhydrazyl (DPPH) was the product of Aladdin Industrial Corporation (D1227007).

### 3.2. Fungus Material

The strain *M. importuna* was collected by Xie Bao-gui (Fungal Research Centre, Fujian Agriculture and Forestry University, Fuzhou, China) and was further identified by the DNA sequence of internal transcribed spacers (ITS) region (Appendix A). The strain has been deposited in College of Life Sciences, Fujian Normal University and has been deposited in the China Centre for Type Culture Collection (CCTCC M 2014324).

### 3.3. Fermentation and Preparation of Extracts

*M. importuna* was cultured by submerged liquid fermentation [55], which was carried out in Erlenmeyer flasks (250 mL) containing 100 mL of potato dextrose medium with a total volume of 40 L. These flasks were incubated at 28 °C with a shaking speed of 180 rpm. The culture broth was centrifuged at 5000 rpm for 30 min to remove the mycelia and then extracted in batches with equal volumes of ethyl acetate. The organic phase was concentrated under reduced pressure by a rotary evaporator to afford the crude extract E1 (3.4 g). Following the same step, another potato dextrose medium which added 1 g bran and 20 g bean sprout juice per liter was used to culture *M. importuna* with a total volume of 36 L and obtained the crude extract E2 (8.8 g).

### 3.4. Isolation and Purification of Compounds ***1**–**10*** [55]

E1 (3.4 g) was firstly separated into three subfractions E1.1 (287.1 mg), E1.2 (107.4 mg), and E1.3 (15.8 mg) by MPLC over RP-C18 silica gel (170 g) using a stepwise gradient CH_3_OH in H_2_O (*v*/*v* 0:100, 5:100, 30:100, 50:50, 70:100, 100:0) based on TLC analyses. Fraction E1.1 was further chromatographed over Sephadex LH-20 (120 g) eluting with methanol and afforded the subfraction E1.11 (20 mg). E1.11 was subjected to silica gel (1.0 g) chromatography using a CHCl_3_-CH_3_OH solvent gradient to yield compound **7** (5 mg). E1.2 was separated by column chromatography over Sephadex LH-20 (120 g) with methanol eluant and afforded the subfractions E1.21 (22 mg). E1.21 was next separated by column chromatography over Sephadex LH-20 (120 g) with acetone eluant and afforded E1.211 (11.4 mg) and E1.212 (7.9 mg). Compounds **5** (5.5 mg) and **6** (5.4 mg) were, respectively, purified from E1.211 and E1.212 by column chromatography over silica gel (1 g) with the eluant of CHCl_3_-CH_3_OH solvent gradient. E1.3 was separated by column chromatography over Sephadex LH-20 (120 g) with methanol eluant and afforded E1.31 (5.4 mg) and E1.32 (7.9 mg). Then, E1.31 and E1.32 were, respectively, subjected to column chromatography over Sephadex LH-20 (120 g) with acetone eluant and yielded pure compounds **4** (2 mg) and **3** (0.7 mg).

Three fractions E2.1 (3.65 g), E2.2 (313.4 mg), and E2.3 (1.6 g) were obtained from the crude extract E2 by MPLC over RP-C18 silica gel (170 g) using 5% and 30% CH_3_OH in H_2_O as the eluant. E2.1 was then subjected to column chromatography over Sephadex LH-20 (120 g) with methanol eluant and afforded subfractions E2.11 (175.4 mg). Compound **10** (4.1 mg) was yielded from E2.11 by column chromatography over silica gel (1 g) eluting with petroleum ether-acetone solvent. E2.2 was then subjected to column chromatography over Sephadex LH-20 (120 g) with methanol eluant and afforded two subfractions E2.21 (236.6 mg) and E2.22 (68.3 mg). Compound **1** (4.2 mg) was yielded from E2.21 by column chromatography over silica gel (1 g) eluting with petroleum ether-acetone solvent. Compound **8** (6.4 mg) was acquired from E2.22 by column chromatography over silica gel (1 g) with the eluant of CHCl_3_-CH_3_OH solvent gradient. Two subfractions E2.31 (37.9 mg) and E2.32 (1.2 g) were afforded from E2.3 by column chromatography over Sephadex LH-20 (120 g) eluting with methanol. Compound **9** (7.4 mg) was purified from E2.31 (37.9 mg) by successive column chromatography over Sephadex LH-20 (120 g) using acetone eluant, and silica gel (1 g) eluting with the chloroform and methanol solution (80:1). The subfraction E2.321 (36.8 mg) was yielded from E2.32 by MPLC over RP-C18 silica gel using 20% CH_3_OH in H_2_O as the eluant. Compound **2** (2.4 mg) was obtained from E2.321 by successive column chromatography over Sephadex LH-20 (120 g) using methanol eluant and silica gel (1 g) eluting with petroleum ether-acetone solvent.

### 3.5. DPPH Free Radical Scavenging Activity

TLC-bioautography [56] was used to determine the free radical scavenging activity of compounds **1**–**10**. These compounds were dispensed to the concentration of 2 mg/mL. Then, 2 μL solution of each compound was spotted onto a precoated silica gel GF254 plate. The TLC plate was left to let the solvent evaporate completely. Then, the plate was sprayed with 0.04% DPPH in ethanol and incubated at 40 °C for 30 min. Any antioxidant compounds could be seen as white spots against the blue background. Ascorbic acid was set as the positive control sample.

The DPPH scavenging rate of these compounds was further assayed by the method reported by Blois [57] and modified by Brand-Williams [58]. Each compound was dissolved in methanol and diluted to 6 different concentrations with the same volume of 94 μL in a 96-well microtiter plate. Then, 20 μL of DPPH dissolved in methanol with a concentration of 0.714 μg/mL was added to these wells. The final concentration of these compounds in each well was 9.8, 19.6, 39.3, 78.6, 157.2, and 314.4 μg/mL, respectively. Ascorbic acid was set as the positive control group with the final concentration of 0.5, 1.0, 2.0, 4.0, 8.0, and 16.0 μg/mL. The reaction mixtures were incubated for 30 min in a dark room at room temperature. The absorbance was read by a microplate reader at 519 nm. The percentage of the DPPH scavenging effect was calculated with the following equation:DPPH scavenging effect (%) = [(A_0_ − A_1_)/A_0_ × 100%] 
where A_1_ is the absorbance of the samples and the standards, and A_0_ is the absorbance of the vehicle.

### 3.6. X-ray Single Crystal Diffraction for Compound ***4*** and ***10***

The single crystal of **4** and **10** was obtained from aqueous acetone. A suitable crystal was selected and subjected to an Oxford Gemini S Ultra diffractometer using Cu-Kα (λ = 1.54184 Å) radiation at 99 K for **4** and using Mo-Kα (λ = 0.7073 Å) radiation at 273 K for **10**. Their structures were determined using the direct method and refined with full-matrix least-squares calculations on *F*^2^ using olex2-2.1 [59]. Crystallographic data have been deposited with the Cambridge Crystallographic Data Centre (CCDC).

## 4. Conclusions

In summary, *M. importuna* can produce abundant phenolics, including two new isobenzofuranone derivatives and one new orsellinaldehyde derivative. These natural metabolites show the significant antioxidant activity of TLC-bioautography and DPPH free radical scavenging assay. The experimental results provided a scientific foundation for the development of the medicinal value of *M. importuna*.

## Figures and Tables

**Figure 1 molecules-28-04760-f001:**
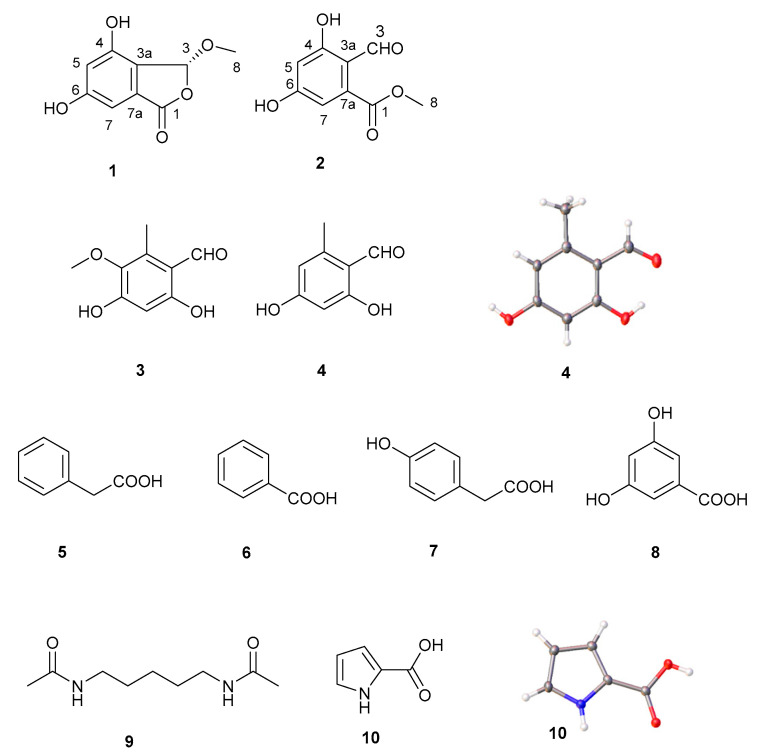
The chemistry structures of compounds **1**–**10** and the crystal form of compound **4** and **10**.

**Figure 2 molecules-28-04760-f002:**
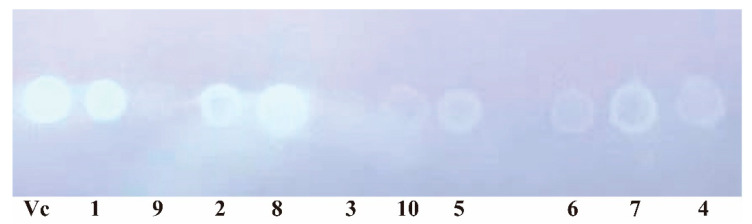
TLC-bioautography test for DPPH radical scavenging activity of compounds **1**–**10**.

**Figure 3 molecules-28-04760-f003:**
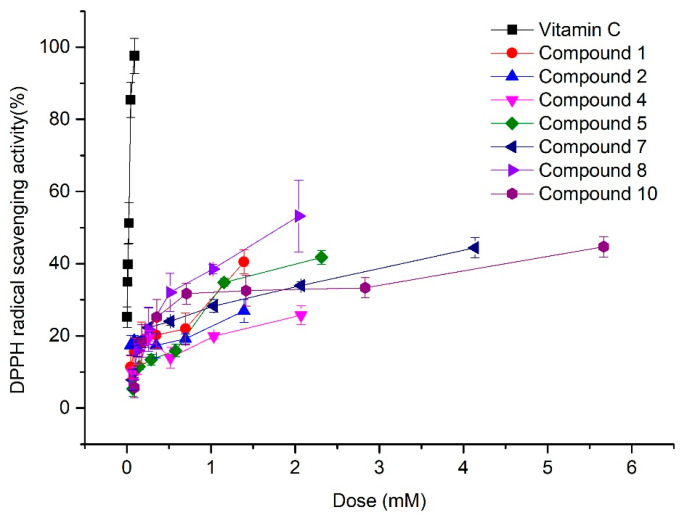
DPPH radical scavenging activity of compounds. (The associated *p* value of SC_50_ of these compounds estimated by using Probit analysis in SPSS 18.0 were less than 0.01, except for **2** with *p* value of 0.064).

**Table 1 molecules-28-04760-t001:** The ^1^H and ^13^C NMR Spectral Data of **1**–**2**.

Position	1		2	
	δ_H_ (Mult, *J* in Hz) *	δ_C_ **	δ_H_ (Mult, *J* in Hz) *	δ_C_ **
1		169.2		167.2
3	6.36 (s, 1H)	103.2	10.33 (s, 1H)	196.6
3a		123.4		113.3
4		155.5		165.4
5	6.68 (d, *J* = 2.0 Hz, 1H)	109.6	6.96 (d, *J* = 2.4 Hz, 1H)	112.1
6		162.5		167.0
7	6.74 (d, *J* = 2.0 Hz, 1H)	103.1	6.52 (d, *J* = 2.4 Hz, 1H)	106.7
7a		131.3		137.8
8	3.49 (s, 3H)	56.2	3.93 (s, 3H)	53.4

* Recorded at 600 MHz in (CD_3_)_2_OD, λ in ppm; ** Recorded at 150 MHz in (CD_3_)_2_OD; λ in ppm.

## Data Availability

The data has supplied in Appendix A.

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
