# Peer review of "DPPH Radical Scavenging Activity of New Phenolics from the Fermentation Broth of Mushroom Morehella importuna"

_molecules, 2023, doi:10.3390/molecules28124760_

Round 1
Reviewer 1 Report
This manuscript represents a deep and thorough study on the composition and useful bioactive activity (Anti-Oxidant Capacity – AOC) of the constituents of mushroom Morehella esculenta. Because of limited disposition of natural mushroom material, the study is carried out in laboratory scale using mycelia. The biomaterial of M. esculenta to be extracted is obtained by the microbial fermentation process (submerged liquid fermentation). Full bio-cycle is employed including cultivation of the culture and obtaining the fermentation broth. Further, the solid is separated from the broth, liquid phase is extracted, the extracts are concentrated, the obtained material is subjected to component isolation. Then, identification is performed and AOC is tested. This research path is fully adequate for achieving the aims of the study.
Generally, the existing information contains information of a high bioactivity of total extracts from the concerned raw material. The main new contribution of this work consists in the isolation of existing and new compounds, identification of their structure and quantification of their antioxidant capacity. In this way, the study supplies a deep look on the individual active components and replies to the question which constituents of the extracts are responsible for their high activity. The obtained new information shows directions for useful applications of the products according to their bioactive qualities.
I don’t find any shortcomings of the paper. It clearly describes in details the work performed. Modern and adequate technique and methods are applied. Specific molecules are identified and their AOC is quantified. My only remark is that many of the abbreviations used are familiar to a very specialized scientific auditory only.
The reported results are useful for prospective applications in pharmaceutical and cosmetic products, may be after development of appropriate technological procedures for production and isolation of the target substances. The latter does not fall in the aims and scope of this study.
The manuscript is fully appropriate by its theme and qualities for publication in the journal Molecules.
Author Response
Thank you very much for your comments. We have revised the manuscript according to your comments. The revisions are addressed point by point below.
1) “HR Q-TOF MS” was changed to “High resolution Quadrupole Time-of-Flight Mass Spectrometry (HR Q-TOF MS)”.
2) “HMBC” was changed to “heteronuclear multiple-bond correlation (HMBC)”.

Reviewer 2 Report
Line 14: Include brief background information on Morehella esculenta for context.
Line 28, 29: To support this statement, it would be beneficial to include a relevant reference
Line 47: Emphasize how M. esculenta mycelia and products from submerged culture are a valid alternative to the fruiting body. Provide relevant references to support your statement.
Line 51: Briefly mention the main conclusions or findings of this study.
Line 162: Including p-values in your data is important to support claims of significant effects.
Line 179: Include the p-value in this figure.
Line 198: From where was this fungus acquired? Please specify the source by indicating the location, origin, and date of collection of the fungus. Additionally, provide the identifying authority responsible for confirming the correct species of the fungus.
Line 201: Provide the reference used for this method.
Line 210: Provide the reference followed for isolation and purification of compounds.
Line 250: Provide the original reference for the DPPH assay. We should honor the pioneering workers for their work.
Line 269: I would like to emphasize that the DPPH assay serves as a preliminary assessment. It is crucial to conduct further research in order to substantiate the claim that the compounds possess medicinal value as antioxidants.
Overall: The DPPH assay serves as a preliminary assessment method for evaluating antioxidant activity. While it offers initial insights into the antioxidant properties of compounds, it is crucial to conduct more comprehensive studies to determine their true potential. Validating the effectiveness and value of compounds claiming to be antioxidants requires further investigation.
Therefore, it is recommended to undertake additional studies to confirm their efficacy and ascertain their worth as antioxidants. This will ensure a more accurate understanding of their capabilities and enable informed decisions regarding their potential medicinal applications.
Author Response
Thank you very much for your suggestions. We have addressed reply in attached file.

Reviewer 3 Report
Dear authors, the article was very relevant in terms of topic and scope of research, however, I request you to add literature review chapter and research results and discussion section.
Author Response
Thank you very much for your suggestions. We have improved the English writing of the entire manuscript in the revised version and addressed them point by point in the attached file.

Reviewer 4 Report
In the manuscript “New natural antioxidants from the fermentation broth of mushroom Morehella esculenta” the authors report the isolation and characterization of new natural oxidants from the fermentation of mushroom Morehella esculenta
The study was interesting but there are some shortcomings:
- An extensive editing and control of punctuation must be improved.
- Line 70 Add the punctuation for the reference 17.
- Line 71 “Rubralide A[17] isolated from 71 Penicillium rubrum showed the dextral optical rotation with the value of +4.0 (c 0.5, EtOH)…” the symbol used for the dextral optical rotation “[α]η” is not clear. Please replace it with the correct notation also in line 51 and 73
- Line 86/87 Report the reference for [Synthetic 87 studies on Sch 202596].
- Line 187 “HR QTOF MS spectra were recorded on Agillent 6520 mass spectrometer in m/z”. Can you report the experimental conditions used for the HR QTOF-MS analysis?
- Line 202/203 How many replicates experiments did you carry out? “M. esculenta was cultured by the submerged liquid fermentation, which was carried out in Erlenmeyer flasks (250 mL) containing 100 mL of potato dextrose medium with a total volume of 40 L”. Did you prepare 40 L of medium for 400 samples? Can you report how did you filter the fermentation broth and the solvent extraction procedure?
- Line 257 Can you write the equation in another form? ( not in jpeg)
Author Response

(The authors gave the same response as above.)

Round 2
Reviewer 2 Report
The title creates a misleading impression of the antioxidant potential since the study solely relied on the DPPH assay to establish antioxidant properties. Therefore, it is crucial to include the DPPH assay in the title.
Author Response
Thank you very much for your suggestions. We have changed the title to “DPPH radical scavenging activity of new phenolics from the fermentation broth of mushroom Morehella importuna”.